# Interventions to Reduce Pesticide Exposure from the Agricultural Sector in Africa: A Workshop Report

**DOI:** 10.3390/ijerph19158973

**Published:** 2022-07-23

**Authors:** Martin Röösli, Samuel Fuhrimann, Aggrey Atuhaire, Hanna-Andrea Rother, James Dabrowski, Brenda Eskenazi, Erik Jørs, Paul C. Jepson, Leslie London, Saloshni Naidoo, Diane S. Rohlman, Ivy Saunyama, Berna van Wendel de Joode, Adeoluwa O. Adeleye, Oyebanji O. Alagbo, Dem Aliaj, Jember Azanaw, Ravichandran Beerappa, Curdin Brugger, Sunisa Chaiklieng, Shala Chetty-Mhlanga, Grace A. Chitra, Venugopal Dhananjayan, Afure Ejomah, Christian Ebere Enyoh, Yamdeu Joseph Hubert Galani, Jonathan N. Hogarh, Janefrances N. Ihedioha, Jeanne Priscille Ingabire, Ellinor Isgren, Yêyinou Laura Estelle Loko, Liana Maree, Nkoum Metou’ou Ernest, Haruna Musa Moda, Edward Mubiru, Mwema Felix Mwema, Immaculate Ndagire, Godwin O. Olutona, Peter Otieno, Jordan M. Paguirigan, Reginald Quansah, Charles Ssemugabo, Seruwo Solomon, Mosudi B. Sosan, Mohammad Bashir Sulaiman, Berhan M. Teklu, Isioma Tongo, Osariyekemwen Uyi, Henry Cueva-Vásquez, Adriana Veludo, Paola Viglietti, Mohamed Aqiel Dalvie

**Affiliations:** 1Swiss Tropical and Public Health Institute (Swiss TPH), 4123 Allschwil, Switzerland; curdin.brugger@swisstph.ch (C.B.); shalachetty7@gmail.com (S.C.-M.); adriana.fernandesveludo@swisstph.ch (A.V.); 2Faculty of Science, University of Basel, 4001 Basel, Switzerland; 3Uganda National Association of Community and Occupational Health (UNACOH), YMCA Building, Plot 37/41, Buganda Road, Kampala P.O. BOX 12590, Uganda; atuagrey3@gmail.com; 4Division of Environmental Health, School of Public Health and Family Medicine, University of Cape Town, Cape Town 7729, South Africa; andrea.rother@uct.ac.za (H.-A.R.); leslie.london@uct.ac.za (L.L.); 5Sustainability Research Unit, Nelson Mandela University, P.O. Box 6531, George 6530, South Africa; james@confluent.co.za; 6Center for Environmental Research and Community Health (CERCH), School of Public Health, University of California, Berkeley, CA 94720, USA; eskenazi@berkeley.edu; 7Odense University Hospital, University of Southern Denmark, 5230 Odense, Denmark; erik.joers@rsyd.dk; 8Oregon IPM Center, Oregon State University, Corvallis, OR 97331, USA; paul.c.jepson@gmail.com; 9Discipline of Public Health Medicine, University of KwaZulu-Natal, Durban 4000, South Africa; naidoos71@ukzn.ac.za; 10College of Public Health, University of Iowa, Iowa City, IA 52242, USA; drohlman@uiowa.edu; 11Food and Agriculture Organization of the United Nations, Subregional Office for Southern Africa, Block 1 Tendeseka Office Park, Eastlea, Harare, Zimbabwe 00153 Rome, Italy; ivy.saunyama@fao.org; 12Infants’ Environmental Health Program (ISA), Central American Institute for Studies on Toxic Substances (IRET), Universidad Nacional de Costa Rica, Heredia 40101, Costa Rica; berendina.vanwendel.dejoode@una.cr; 13Department of Crop Production and Protection, Obafemi Awolowo University, Ile-Ife 220282, Nigeria; adeleyeadeoluwa@gmail.com (A.O.A.); olubunmialagbo@gmail.com (O.O.A.); msosan@oauife.edu.ng (M.B.S.); 14Department of Health Sciences and Medicine, University of Lucerne, 6002 Lucerne, Switzerland; dem-aliaj@hotmail.com; 15Department of Environmental and Occupational Health and Safety, Institute of Public Health, College of Medicine and Health Sciences, University of Gondar, Gondar, Ethiopia; jemberazanaw21@gmail.com; 16ICMR-Regional Occupational Health Centre (Southern), Bangalore 562110, India; ravichandranrohc@gmail.com (R.B.); dhananjayan_v@yahoo.com (V.D.); 17Department of Environmental Health, Occupational Health and Safety, Faculty of Public Health, Khon Kaen University, Khon Kaen 40002, Thailand; csunis@kku.ac.th; 18Global Institute of Public Health, Ananthapuri Hospitals and Research Institute, Trivandrum 695024, Kerala, India; achitragrace@gmail.com; 19Department of Animal and Environmental Biology, University of Benin, P.M.B. 1154, Benin City 300212, Nigeria; afureejomah@gmail.com (A.E.); osariyekemwen.uyi@uniben.edu (O.U.); 20Green and Sustainable Chemical Technologies, Graduate School of Science and Engineering, Saitama University, Saitama 3388570, Japan; enyoh.c.e.527@ms.saitama-u.ac.jp; 21Section of Natural and Applied Sciences, School of Psychology and Life Sciences, Canterbury Christ Church University, Canterbury CT1 1QU, UK; joseph.galaniyamdeu@canterbury.ac.uk; 22Department of Environmental Science, College of Science, Kwame Nkrumah University of Science and Technology, Kumasi, Ghana; jhogarh@gmail.com; 23Department of Pure and Industrial Chemistry, University of Nigeria, Nsukka 410001, Nigeria; janefrances.ihedioha@unn.edu.ng (J.N.I.); sulaimanmuhammadbashir@gmail.com (M.B.S.); 24Horticulture Program, Rwanda Agriculture and Animal Resources Development Board, Kigali 5016, Rwanda; kijpriscile@gmail.com; 25Lund University Centre for Sustainability Studies (LUCSUS), P.O. Box 170, SE-221 00 Lund, Sweden; ellinor.isgren@lucsus.lu.se; 26Ecole Nationale Supérieure des Biosciences et Biotechnologies Appliquées (ENSBBA), Université Nationale des Sciences, Technologies, Ingénierie et Mathématiques (UNSTIM), BP 2282 Abomey, Benin; lokoestelle@yahoo.fr; 27Department of Medical Bioscience, University of the Western Cape, Bellville 7493, South Africa; lmaree@uwc.ac.za; 28Ministry of Agriculture and Rural Development, Cameroon, Direction of Regulation and Quality Control of Agricultural Inputs and Product, Messa, Yaoundé P.O. Box 2082, Cameroon; nkoumernest@gmail.com; 29Department of Health Professions, Manchester Metropolitan University, Manchester M15 6BG, UK; h.moda@mmu.ac.uk; 30Chemistry Department, School of Physical Sciences, College of Natural Sciences, Makerere University, Kampala, Uganda; mubirued@gmail.com; 31School of Materials, Energy, Water and Environmental Sciences, The Nelson Mandela African Institution of Science and Technology, Arusha P.O. Box 447, Tanzania; mwema.felix@nm-aist.ac.tz; 32Southern and Eastern Africa Trade Information and Negotiation Institute (SEATINI) Uganda, Kampala P.O. Box 3138, Uganda; indagire@seatiniuganda.org; 33Industrial Chemistry Programme, College of Agriculture Engineering and Science, Bowen University, Iwo 232101, Nigeria; oladele.olutona@bowen.edu.ng; 34Pest Control Products Board, Loresho, Nairobi P.O. Box 13794-00800, Kenya; p.otieno@pcpb.go.ke; 35Common Services Laboratory, Food and Drug Administration (FDA) Philippines, Alabang, Muntinlupa 1781, Philippines; jmpaguiriganrph@gmail.com; 36School of Public Health, University of Ghana, Accra P.O. Box LG13, Ghana; rquansah@ug.edu.gh; 37Department of Disease Control and Environmental Health, School of Public Health, Makerere University College of Health Sciences, Kampala P.O. Box 7072, Uganda; cssemugabo@musph.ac.ug; 38CropLife Uganda, Chicken House, Plot1, Old Kampala Road, Second Floor Room 17, Kampala P.O. Box 36592, Uganda; seruwosolomon@yahoo.com; 39Ethiopian Agriculture Authority, Addis Ababa P.O. Box 313003, Ethiopia; teklume@itu.edu.tr; 40Faculty of Naval and Ocean Engineering, Istanbul Technical University, Maslak P.O. Box 34469, Turkey; 41Laboratory for Ecotoxicology and Environmental Forensics, Department of Animal and Environmental Biology, University of Benin, P.M.B. 1154, Benin City 300212, Nigeria; isioma.tongo@uniben.edu; 42Department of Zoology and Entomology, Faculty of Natural and Agricultural Sciences, University of the Free State, P.O. Box 339, Bloemfontein 9300, South Africa; 43Facultad de Ciencias de la Salud, Carrera de Medicina Humana Lima, Universidad Científica del Sur, Lima 15067, Peru; henry.cueva.md@gmail.com; 44Centre for Environmental and Occupational Health (CEOHR), School of Public Health and Family Medicine, University of Cape Town, Cape Town 7700, South Africa; paolavig@gmail.com

**Keywords:** pesticides, risk assessment, Africa, sub-Saharan Africa, environmental health, occupational health, interventions, evidence-based policymaking, personal protective equipment, stakeholders, mixed methods, integrated pest management (IPM)

## Abstract

Despite the fact that several cases of unsafe pesticide use among farmers in different parts of Africa have been documented, there is limited evidence regarding which specific interventions are effective in reducing pesticide exposure and associated risks to human health and ecology. The overall goal of the African Pesticide Intervention Project (APsent) study is to better understand ongoing research and public health activities related to interventions in Africa through the implementation of suitable target-specific situations or use contexts. A systematic review of the scientific literature on pesticide intervention studies with a focus on Africa was conducted. This was followed by a qualitative survey among stakeholders involved in pesticide research or management in the African region to learn about barriers to and promoters of successful interventions. The project was concluded with an international workshop in November 2021, where a broad range of topics relevant to occupational and environmental health risks were discussed such as acute poisoning, street pesticides, switching to alternatives, or disposal of empty pesticide containers. Key areas of improvement identified were training on pesticide usage techniques, research on the effectiveness of interventions targeted at exposure reduction and/or behavioral changes, awareness raising, implementation of adequate policies, and enforcement of regulations and processes.

## 1. Introduction

The growing world population continues to put a strain on the agricultural sector and its need for effective and innovative methods of production to ensure food and job security through sufficient yields in crop production. Agriculture is, therefore, a primary sector in most countries, especially in sub-Saharan Africa (SSA), where most of the countries are categorized as low- and middle-income countries (LMICs). In these countries, most citizens rely on small-scale agriculture for food or income; there is a low tax base, and governments are dependent on international funding for projects [1]. Innovative and strategic international partnerships have recently emerged with the goal of sustainable agriculture in Africa to increase income and improve the food security of 30 million smallholder farm households in five agricultural hot spots [2]. In parallel, there is emerging evidence of the impact of climate change on social and economic development due to the fact of reduced agricultural productivity [3]. Unexpected recent plagues, for example, desert locust (*Schistocerca gregaria*) or the fall armyworm (*Spodoptera frugiperda*), continue to threaten livelihoods in the agricultural sector in the Global South [4]. These major threats to food supply continue to demand effective pest management strategies, which often include the use of synthetic pesticides. Many pesticides have been classified as persistent and highly hazardous to the environment and human health (e.g., pesticides in the chemical groups of organochlorines, organophosphates, or carbamates). Numerous attempts have been made by international organizations to address these global public health concerns for health and the environment with policies and regulation enforcement including the identification and labeling of highly hazardous pesticides (HHPs) and its categories of harm to health and the environment [5]. Yet, despite these attempts, the global situation of pesticide management reported by the Food and Agriculture Organization of the United Nations (FAO) and the World Health Organization’s (WHO) survey [6] revealed that one-third of the countries they investigated did not have guidelines for HHP use, posing the problem of evaluating products without guidelines. Studies on the knowledge attitudes and practices (KAP) of farmers conclude that insufficient training exists on safe pesticide use practices [7,8] and show noncompliance in the use of personal protective equipment (PPE), resulting from the influence of workers’ socio-cultural context (i.e., gender dynamics and social status), herbicide risk perceptions, and working conditions (i.e., environmental and logistical) [9]. The use of simple models to quantify the risks associated with pesticides to nontarget endpoints before and after registration is also lacking in many African countries [10,11]. There are also concerns regarding exposure for bystanders and farming communities including children engaged in farming activities [12,13,14]. Dietary exposure due to the high concentrations of pesticide residues in food has also been reported in the African region [15,16,17,18,19,20,21,22,23,24].

These conclusions from the literature raise concerns that there is no effective strategy in place to support the diverse and challenging situations in the context of pesticide use, which places the burden of risk largely on poor smallholder farmers and farm workers. An effective strategy may be defined as an activity or set of activities aimed at modifying a process, course of action, or sequence of events in order to change one or several of their characteristics such as the performance of the expected outcome. This may include educational programs, regulation, development and enforcement of new or more stringent policies, other improvements in the environment, or public health promotion campaigns. Interventions that include multiple strategies and a multiplicity of actors—public, private, and civil society—are typically the most effective in producing desired and long-term outcomes. The evidence has shown that interventions create change by (i) influencing individuals’ knowledge, attitudes, beliefs, and skills; (ii) increasing social support; (iii) creating supportive environments, policies, and resources, in other words, promoting a culture of human and environmental health protection [25].

This paper presents the key issues and recommendations raised in a workshop with global pesticide experts regarding their perceptions of the most pressing issues and effective interventions that are contextual for the agricultural sector in Africa and have the potential to improve the situation for the environment and human health.

## 2. Materials and Methods

### 2.1. Workshop Preparation

To prepare for the workshop, two main activities were conducted. First, a systematic literature review was conducted to map current geographical research hot spots and identify gaps around environmental and public health risks associated with agriculture pesticide use in the Africa Region [26]. In this review, 391 articles published between 2006 and 2021 were identified that covered 469 study sites with five geographical research hot spots (two in South Africa, two in East Africa, and one in West Africa). The systematic review identified key researchers to invite to participate and important pest problems, research hot spots, and interventions implemented for discussion at the workshop.

Second, a mixed-method content analysis study was conducted. We combined an online survey with 36 stakeholders from 16 different countries, one-to-one interviews with two individuals, and a closed focus group discussion with five relevant researchers and stakeholders in the African agricultural sector. Stakeholders for this survey were identified through the Pesticide Forum Network (PDF) led by the Environmental Division, School of Public Health and Family Medicine, University of Cape Town (UCT), and from the author list of papers identified in the systematic review of pesticide research in the Africa Region.

Based on this information, an online workshop was prepared, scheduled, and successfully held on 15–17 November 2021. The workshop aimed to discuss possible interventions to reduce pesticide exposure in the agricultural sector in Africa. We invited 14 speakers who had also participated in the mixed-method survey. On the first day, we discussed these targets focusing on occupational settings and exposure reduction among workers. On the second day, we primarily addressed interventions related to reducing environmental and public pesticide exposure. On the third day, we focused on policy measures. Each day started with introductory talks followed by a selection of case study presentations.

The workshop delivery approach included a mix of presentations, breakout group discussions, and plenary discussions. To stimulate the breakout group discussion, we started the discussion with polls addressing the main group’s discussion topics. Then, a moderated group discussion took place. Conclusions were shared in a subsequent plenary. The minutes from each session were used for writing this paper. 

### 2.2. Workshop Participants

Participation in the workshop was without cost and open to anybody who was interested. Information regarding the workshop was shared publicly through different online platforms, directly with the participants of the mixed-method survey, and within the organizers’ networks. In total, 198 participants from 37 countries registered for the meeting; of those, 53 registered for one or two days only. Most of the participants were from Africa (67%), followed by Europe (18%), America (9%), and Asia (4%) (Figure 1). Most of the participants declared to be researchers (69%), followed by representatives of authorities (10%), nongovernmental organizations (7%), practitioners (4%), and industry (3%). The duration of professional experience with pesticides varied over a wide range (Figure 2). 

## 3. Results

The workshop presentations and recordings are available at https://www.swisstph.ch/en/about/events/interventions-to-reduce-pesticide-exposure-from-agriculture-sector-in-africa/, accessed on 18 July 2022.

To start the workshop, Samuel Fuhrimann gave an overview regarding the review of the pesticide research literature [9]. In total, 391 articles were identified that described data from 469 study sites. Human risks were addressed in 49% of the study sites and environmental risks in 20%. In many papers addressing human health or environmental risks, environmental samples were analyzed (68%). At a total of 180 (38%) study sites, human subjects were investigated. However, prospective longitudinal studies were only conducted at 18 sites (10%), and interventions on how to reduce pesticide use were only addressed at four sites (2%). High-quality research studies (e.g., large prospective cohort studies or randomized controlled trials) to monitor the (cost)-effectivity of the implemented interventions were missing. This clearly demonstrates the lack of high-quality research, which is needed for evidence-based policy interventions. 

Martin Röösli presented the results of the stakeholder survey conducted before the workshop. When asked about the top guiding documents recommended to reduce agriculture pesticide use and its harm to the environment and human health, the following documents were most often named: The International Code of Conduct on Pesticide Management [27], FAO Pesticide Registration Toolkit [28], the WHO Recommended Classification of Hazardous Pesticides [29], and identifying HHPs using FAO and WHO Joint Meeting on Pesticide Management (JMPM) criteria [30]. From the content analysis of the survey, four different targets for interventions aiming at reducing pesticide use were identified: training, research, awareness raising, and policies/regulations. The stakeholders considered the fiscal and multiple economic implications as the main challenges for the implementation of policies to enforce reduced pesticide use. It was emphasized that more focus should be put on the implementation of science in order to bridge the gap between research and efficient implementation.

### 3.1. Occupational Pesticide Exposure

#### 3.1.1. Problems in the Occupational Setting

Occupational exposure to pesticides among farmers usually occurs in agricultural fields directly in contact with pesticides during spraying and mixing. They become the most vulnerable group to pesticide exposure [31]. Additionally, events such as accidental spills, splashes, and consumption by mistake may result in acute poisonings [32]. In LMIC occupational settings, acute poisonings are one of the most pertinent problems, which was addressed by Erik Jørs. Globally, it is estimated that approximately 385 million cases of unintentional pesticide poisoning occur annually, including approximately 11,000 occupational fatalities [33,34]. The number of such cases is difficult to estimate due to the fact of underreporting and challenging diagnosis. The main reasons for accidental pesticide poisoning are increased accessibility and availability of HHPs; inadequate availability and usage of PPE such as clothes/overalls, shoes, and masks; inadequate washing of pesticide contamination; improper storage; illegal street sellers of pesticides. Sapbamrer et al. (2020) [35] found that the determinants associated with the use of PPE and pesticide safety practices were demographic factors (i.e., education/literacy level, experience of illness, and income); farm structure factors; behavioral and psychosocial factors (i.e., perceptions, attitudes, awareness, norms, and beliefs); training-related factors (i.e., information on pesticides, access to extension services, and training programs). Therefore, it is important to educate this group of workers to change their perceptions and behavior toward safe pesticide use and handling. Among the consequences of acute poisonings are reduced cholinesterase activities, DNA damage [36], and other health complaints [31,36,37,38]. In addition, pesticides are used for suicides, which are most effectively prevented by reduced accessibility [39]. 

Within the framework of IPM, Paul Jepson discussed the selection of pesticides for reducing human health and environmental risks using the example of the fall armyworm. Farmers lack access to basic information on pest biology and life cycle, efficacious control methods, and application timing and safer use of pesticides, which is exacerbated by limited access to extension services. Agrochemical dealers, who could provide some guidance in selecting lower-risk pesticides, have limited access to training and professional development other than via industry seminars and may, thus, miss out on critical information regarding integrated pest management (IPM). Both farmers and agrochemical dealers lacked PPE onsite [40]. Paul Jepson presented a system to classify 659 pesticides with respect to human and environmental health [41]. A standalone guideline, included within this publication, allows farmers to select lower-risk pesticides to protect applicators, human bystanders, aquatic life, terrestrial wildlife, and pollinators. The system is already in use among millions of certified farms internationally. One practical challenge, however, is that low-risk pesticides are often more costly than HHPs or other high-risk chemicals. This work was based upon an extensive multiscale analysis of pesticide risks in five West African countries [42], which demonstrate some of the highest health and environmental risks ever published.

#### 3.1.2. Case Studies

Insufficient pesticide-related knowledge, attitude, and practices among smallholder farmers and retailers have been documented in Uganda [43,44,45,46], Nigeria [47], and other African countries [48]. As a case study, Aggrey Atuhaire presented a 12 month randomized control trial (October 2020–October 2021) on the effect of targeted information access (i.e., training and SMS reception) on the responsible use and handling of pesticides among smallholder farmers in Uganda (APSENT-Uganda). The study built upon evidence generated over the past six years within the PESTROP-Uganda project, which generated an interdisciplinary evidence base on environmental and public health issues of smallholder farmers in Uganda [44,45,49,50]. The intervention study included a 2 day training workshop on the responsible use of pesticides and a 5 month mobile phone SMS structured campaign on the theme of personal protective clothing, targeting 360 and 180 conventional smallholder farmers, respectively, who had previously been assessed through a baseline survey. Preliminary results comparing baseline and follow-up data indicated that farmers’ knowledge and attitude improvement were not necessarily reflected in their practice. Nevertheless, the results showed notable improvements in certain practices, especially farmers buying and using certain protective clothing such as waterproof pants, long-sleeved jackets, and chemical-resistant gloves. With regards to improvement, knowledge and interpretation of pesticide label pictograms was a key area of high performance for the majority of farmers. Up to 150 products and 50 active ingredients, the majority of which are insecticides and WHO hazard class II pesticides, were found to be used during this time; changes in pesticide use (i.e, toxicity, application frequency, amount, and area) are still being explored. Aggrey Atuhaire pointed out that results of preliminary analyses suggest that information dissemination to farmers shows great potential, but a more holistic approach, beyond information access, is needed to sustainably improve sound management of pesticides among smallholder farmers from an LMIC context. One main catalyst of pesticide awareness in the future would be to leverage the rapidly expanding telecommunication system (such as mobile phones) to reach out to farmers in their own language. This could involve developing a mobile app and transcribing label information to different local languages [51]. 

Diane Rohlman presented an intervention study focused on changing risk perceptions and behaviors among Egyptian adolescent pesticide applicators. Previous work with this population identified behaviors that increased pesticide exposure (e.g., mixing pesticides with a stick and hygiene behaviors) [52]. Focus groups were used to share this information with the adolescent applicators, their parents, and the Ministry of Agriculture to learn their perspectives and identify feasible ways to reduce exposure. Based on this qualitative research, an educational intervention was developed that targeted three behaviors: using a stick instead of their hand to mix pesticides, minimizing walking in the spray, and bathing and wearing clean clothes [53]. The study found increased awareness of the hazards of pesticides, changes in attitudes, and significant improvements in PPE use after the intervention. Participatory approaches are considered to be the most useful for developing feasible and acceptable interventions. 

#### 3.1.3. Discussion of Promoters and Barriers to Switching to Alternatives

Before the breakout group discussion, workshop participants were asked about the most relevant occupational risks of pesticide applicators. Exposure to pesticides was mentioned most often, followed by injuries and musculoskeletal diseases (Figure 3). Infectious diseases, dust, and psychological distress were considered less relevant. For the initiation of research projects, international networks with funding and local research or university initiatives were considered to be most helpful. When presented with a list of potential topics for occupational health research, all items received high priority (Figure 3).

The topics of these polls were subsequently discussed in depth. In terms of health risks to applicators, it was emphasized that the risk perception of pesticide applicators and farmers is an important aspect. Sometimes, it is challenging to convince farmers that pesticides are a health risk. Only listing health symptoms is usually not sufficient, as farmers need to understand the route of exposure and subsequent acute and chronic health effects. In particular, raising awareness of the potential long-term risks and accompanying long-term economic implications if applicators become ill is a challenge. These hidden costs are less evident to farmers than the short economic benefits of applying pesticides. One also needs to consider cultural issues that may prevent the application of correct protection measures such as “you are not a man if you use protection” or “we have done this for years like this”. Anecdotal reports suggest that the COVID-19 pandemic had an impact on attitudes toward mask wearing. Knowledge transfer to the target population is thus considered to be of high priority. Much of the (important) information is not available in local languages, which is a common barrier for effective communication. In the absence of language skills, red–green color blindness is also a barrier for correctly understanding pictograms. Communication should be “two-way” and include a participatory component to be effective. Collaborating with peer educators was found to work well. 

For initiating research projects, raising awareness was considered one of the key triggers. Collaboration with NGOs may be a way to make local politicians aware of the situation. Research projects are often small and have limited reach, although potentially useful for sensitizing the local population. To be effective, one should not look at pesticides in an isolated manner but address them together with other important public health problems, such as reproduction and HIV, or occupational health outcomes such as musculoskeletal diseases and psychological distress. This may allow for larger scale and sustainable interventions by profiting from the mutual co-benefits of the interventions. In terms of international funding, raising awareness at the level of funding bodies is needed. To date, pesticide exposure has relatively little priority. International researchers and stakeholder networks may help break the vicious cycle of little research money, which results in problems with awareness. 

Figure 4 shows that various topics were considered to be of priority for future research. Appropriate attitudes on handling pesticides and the health effects of pesticides obtained a slightly higher priority compared to the other suggestions. In the subsequent discussion, the need for ecotoxicological research and biomonitoring of applicators and the public was also emphasized. Bottom-up approaches, such as first identifying the knowledge gaps of workers, were considered important for effective research. In terms of health effects, acute poisonings and the effects of long-term exposure to low levels were considered to be the most pertinent. Even if uncertainties remain regarding the toxicity of various compounds, research on the most effective prevention was of high relevance for farmers in addition to toxicological research. 

Subsequently, the barriers (Figure 5), promoters (Figure 6), and trade-offs (Figure 7) when switching to alternatives (i.e., reduced pesticide use or organic farming) were discussed. Several participants suggested that barriers received a high degree of consent, such as lack of knowledge, financial pressure, lack of alternatives, risk perception of workers, and lack of insurance for harvest loss. It was also mentioned that low-risk pesticides were more expensive. Several proposed promoters were considered to be relevant for a transformation to low-risk agricultural methods (Figure 6). In terms of trade-offs, stress, insecurity, and income loss were considered the most relevant, whereas an increase in workload and job security obtained little consent from the workshop participants (Figure 7). 

In the discussion, it was emphasized that the size of a farm is critical for transformation processes. A small family farm may be less affected by market factors than a large employer. If a farm is producing for export, they need to adhere to international standards. Thus, international standards are important promoters of transition. Change in consumers’ preferences may eventually also result in pressure to change to alternatives. As a comparison, plastic packaging in supermarkets was reduced due to the pressure from consumers. A transition from conventional farming to alternatives may result in immediate income losses, and farmers need confidence that in the long term this will be compensated with benefits and better environmental conditions. However, it was also questioned to what extent transition really causes income loss. Well-trained farmers may know about working alternatives and saving expenses on pesticides. Thus, profitability may even increase in certain circumstances. 

### 3.2. Environmental and Public Pesticide Exposure

#### 3.2.1. Problems for the Environment and Public Exposure

On the 2nd day of the workshop, James M. Dabrowski introduced the development of decision support tools for managing pesticide risks to aquatic ecosystems and human health in South African water resources [54,55]. Several indicators, such as crop type, growth stage, and meteorological and geographic factors, were tested in the field and could reliably predict relative differences in pesticide concentrations. Simple modeling approaches were reliable and should be used more widely. These novel tools are thus useful for identifying hot spots across a country, prioritizing pesticides based on their risk to human health and the aquatic ecosystem, identifying important transport routes, and for informing which pesticides should be included in monitoring programs. 

Aqiel Dalvie discussed lessons from the fields with respect to the implications of interventions. In his studies, he found that sprayers, non-sprayers, farm residents, and neighboring nonfarm residents, including vulnerable groups, were at risk for exposure and possible long-term adverse health effects [12,56,57,58,59,60,61,62,63,64,65]. Proximity is a relevant factor for aerial exposure, but other sources of exposure are also relevant such as drinking water from an open water source, eating crops from the agricultural areas and gardens, or obsolete stocks. Interventions may target the re-entry of workers into orchards/vineyards, PPE for sprayers and non-sprayers, reduction in pesticide usage and alternatives, buffer zones, awareness and education, surveillance, improved legislation and implementation, improved access to health care, and screening and conducting biomonitoring.

Godwin O. Olutona presented their findings of higher levels of organochlorine pesticides (OCPs) above the recommended limit in water and bottom sediment of Aiba reservoir, Iwo [66], as well as in leguminous food crops from selected markets in Ibadan, Nigeria [67]. In addition, an incident was presented concerning 116 students from a secondary school in Doma, Gombe State, and 112 people in Cross River (Nigeria), who fell ill and were hospitalized after eating cowpea contaminated with pesticides [68]. Twenty fast food outlets were closed in Nigeria because of fatalities traced to pesticide residue in their products [69]. Pesticides residues were also detected in malt drinks sold in Nigeria market, which was traced to the pesticide residue used in treating sorghum in 2018 [70].

Samuel Fuhrimann presented the results from a series of recent pesticide exposure assessment studies. An analysis of 27 currently used pesticides at 20 air sampling sites across Africa over seven years demonstrated the ongoing use of pesticides in Africa that are banned in HICs [71]. For epidemiological research and risk management, good knowledge of the exposure situation is needed. Presented examples included seasonal variations in air concentrations of 27 organochlorine pesticides (OCPs) and 25 current-use pesticides (CUPs) [72], the water concentrations of 53 pesticides in catchments across three agricultural areas of South Africa [73], and in-depth studies on personal exposure using silicon wristbands for children and guardians during spraying season [74,75]. From concurrently sampled air and soil samples, it was concluded that, except for chlorpyrifos, soil ingestion generally represented a minor exposure pathway compared to inhalation (i.e., <5%) [76]. He suggested that a pesticide-vigilance system should be introduced, i.e., a post-registration monitoring analogue to current practice for pharmaceutics (pharmacovigilance).

Brenda Eskenazi presented lessons from more than 20 years of research with the Center for the Health Assessment of Mothers and Children of Salinas (CHAMACOS), which is a community–university partnership. Between 1999 and 2000, pregnant mothers were recruited and followed up. The study showed that mothers had higher organophosphate exposure than the US average, and higher levels were associated with various negative effects on the cognitive development and respiratory health of their children [77]. Effects from exposure during pregnancy were still detectable in adolescence. In the subsequent CHAMACOS study, interventions were explored to reduce pesticide exposure. As an example, regarding how different health risks are interrelated, Brenda Eskenazi demonstrated that Californian farm workers were at increased risk of COVID-19 during the pandemic.

#### 3.2.2. Case Studies

Saloshni Naidoo presented various studies on pesticide safety practices amongst small-scale women farmers in KwaZulu-Natal, South Africa. It was initially found that pesticides were stored in the open or even at home, and only a minority locked up pesticides. Handling and re-use of empty pesticide containers also occurred regularly and, thus, is a source of pesticide exposure [78]. After a two-day training program, the situation markedly improved. Access to services was one of the health system challenges identified, as emergency services are not available or far away. Furthermore, the operating hours of some services are not compatible with working hours. There was also limited awareness in the health system of pesticide-related health problems as demonstrated in a case study of acute chlorpyrifos poisoning in pregnancy [79]. 

Berna van Wendel de Joode presented several tested approaches to reduce occupational and environmental pesticide exposures in Costa Rica embedded in the Infants’ Environmental Health Program (ISA) [80,81]. It was found that communication of study results with the study population, ministries, and industry resulted in significant improvements. By collaboration with key actors, exposures from drift could be successfully reduced. Flexible educational strategies were designed to stimulate social and cognitive development. Utilizing a participatory approach, agro-ecological concepts were discussed and experienced during workshops in the field. 

#### 3.2.3. Discussion of Interventions

The subsequent discussion was opened by a poll regarding the most relevant exposure pathways for health. Nutrition (i.e., ingestion of pesticide residues in food) was considered to be most relevant followed by drinking water and air including drift and dermal contact (Figure 8). 

In terms of ecological research priorities, biodiversity and the effect of pesticides on aquatic organisms received the highest score, although the differences were small for the suggested topics (Figure 9). Group discussion revealed pest resistance to pesticides and the effects of pesticides on microbial activity as areas where further research is needed. A challenge in many African countries is the limited number of laboratories with competencies for analysis of environmental samples and the high costs for these types of analyses. This hampers progress in research and in identifying the most critical situations. For health effects research, vulnerable populations were considered a key priority (Figure 9). 

There was a relatively high conviction that making alternatives well known is effective for reducing pesticide levels in the environment (Figure 10). Switching to less hazardous substances and interventions regarding awareness paralleled with the KAP of workers were also considered effective. Nevertheless, additional research on the effectiveness of various interventions was deemed to be important (Figure 11). The highest priority was given to interventions addressing the switch to less hazardous substances and the KAP of workers as well as the general population. Similar to the discussion on occupational research and interventions presented above, participatory approaches with the public were considered the most important. This may also be labeled as citizen science or MARP (“Méthode Active de Recherche Participative”). Participatory approaches may be used for defining research questions, acceptable interventions, and data collection. If the public is involved in data collection, the results need to be communicated to them. The information for the public should not be very technical and not too complicated. Communicating environmental pesticides levels may result in higher awareness and opposition against pesticides. It was noted that in many LMICs, people were mostly focused on food security and less on food safety. To change this mindset, comprehensive data regarding ecological and human health risks are needed. Local data are usually much more convincing than data collected from other countries, even if they would be transferable. 

In addition to specific interventions, the need for system changes was emphasized. Agriculture production should move away from monoculture farming, since emergency outbreaks are more common in such a setting. In the case of an emergency (e.g., fall army-worm), there is hardly any alternative to pesticide application in order not to lose the harvest.

It was further discussed that educating the next generation of safety professionals and children may be a sustainable intervention with long-term consequences for the agricultural system. They may influence their parents’ decisions on whether and how to apply pesticides. Conversely, the training of farmers did not always result in a change in behavior.

### 3.3. Policy Measures

#### 3.3.1. Evidence Needs from a Stakeholder Perspective

Ivy Saunyama discussed what types of evidence are needed from intervention studies for the regulation of pesticides to reduce exposure from the agricultural sector. For the regulator, the most critical evidence is the potential benefits, in terms of minimizing crop losses, reduction of diseases, or protection of buildings, that outweigh the expected risks to applicators, consumers, and the environment. Of note, HHPs are causing the majority of environmental and health problems. The need to take action on HHPs is widely recognized and long-awaited. 

#### 3.3.2. Illegal and Street Pesticides

Hanna-Andrea Rother used the example of street pesticides to demonstrate that exposure risks from agricultural pesticides are not limited to crop protection. Street pesticides are either illegal pesticides legally registered for agricultural use and then decanted into unlabeled domestic containers, such as juice bottles, or unregistered prepackaged products [82]. The former poses a high risk for accidental ingestion (e.g., children), but street pesticides may also be the subject of homicides and suicides [83,84,85]. Poverty and housing inequality are key reasons for pest infestation and, thus, interventions need to simultaneously address issues at the individual, community, and regulatory levels to change the context. Regulatory interventions are needed to reduce access to agricultural pesticides by suppliers and consumers. Training and risk communication tools need to address different target groups ranging from street vendors to children and policymakers [86]. For exposure reduction, a broad range of methods are needed using multipronged approaches, including codesigning interventions with the target audience. However, to reduce hazardous exposures to agricultural pesticides in urban settings, firstly, HHPs (especially those banned in the Global North) should be phased out from agricultural use and alternatives implemented [83]. 

#### 3.3.3. What Policy Is Needed?

Leslie London discussed policy responses to reduce occupational and environmental exposure to pesticides in Africa [87]. Policy is important for import and distribution, usage, regulation, surveillance, and research. Approximately 10% of global imports was by Africa, which corresponds to USD 3 billion, in 2021. Despite rapid growth in pesticide markets, policy enforcement has not kept pace, resulting in fake and counterfeit pesticides, negative environmental impacts, and harm to health. In terms of usage, climate shocks were found to drive food insecurity with the subsequent need of pesticides for buffering. Whereas a lack of risk awareness and peer pressure were responsible for part of the unsafe use, as fairly often farmers were not aware of the risk, but PPE is too expensive or unavailable, and alternatives to chemical control are not available or promoted by government extension agents. To address peer pressure, the promotion of Farmer Field Schools as the mainstream is important. Regulations for retailers and for advertising are often lacking or, if existing, not enforced. Policies that control who purchases and what volumes can be sold, phasing out hidden subsidies for pesticide inputs, and actively reducing sole reliance on chemicals for pest control must supplement strategies reliant on farmer behavior.

#### 3.3.4. Discussion of Stakeholder Perspectives and Policy Options

In the subsequent discussion sessions, participants voted on the priority measures to be implemented (Figure 12). The three most commonly selected measures were to improve the knowledge, attitude and practice (KAP) of workers (farmer field school), more research on health effects (hazard potential), and more intervention research on organic alternatives. Strikingly, a majority of participants were convinced that insufficient implementation and enforcement of policies were the most pertinent problems but not the lack of policies (Figure 13). Nevertheless, some participants were surprised to see that many people thought that there were enough policies. Specifically, it was mentioned that incentives were missing to promote less pesticide use. 

A lack of resources and research were identified as barriers to faster transition to less hazardous practices. In this context, the responsibility of the pesticide manufacturers and retailers were discussed. In principle, parts of the profit from industry could be used for research. However, this is very sensitive in terms of conflict of interests and would need the careful implementation of firewalls so that selection of research questions, research groups, and research projects is not influenced by the donor. Ideally, such a research fund should be managed by the government. 

As a typical example of an important problem, inadequate disposal of obsolete pesticides and empty pesticide containers was mentioned a few times during the workshop. The reasons for this problem are unclear or lack of responsibilities. Often, adequate disposal in practice becomes the personal responsibility of the user (i.e., farmer), which is not ideal according to the workshop participants. There was no clear preference on who should be mainly responsible (Figure 13). 

It was suggested that a pesticide vigilance system similar to a pharmacovigilance system would be helpful in identifying the most relevant emerging problems and reacting to them. Some examples of such locally applied surveillance systems were mentioned. Whereas the methods are clear for acute poisoning incidents, such a system is more challenging for capturing long-term and chronic effects.

## 4. Discussion

To our knowledge, this workshop was the first of its kind in Africa. Overall, a broad range of topics relevant to occupational and environmental health risks and interventions related to a reduction in pesticide use were discussed during the workshop. A lack of intervention and longitudinal studies in the region was identified in the systematic literature review. An integrated strategy to reduce pesticide use includes awareness-raising, training of different populations, and research as well as the implementation and enforcement of policies and regulations. Participatory approaches that include the concerned populations are most effective. In occupational settings, the main barriers for switching to less hazardous alternatives are high costs, lack of awareness, paucity of experiences and knowledge regarding alternatives for specific applications, and a lack of insurance for harvest loss. A lack of resources is the main reason for many of the deficits in this area, most likely due to the fact of inadequate financing and not accounting for the long-term costs related to the negative effects of pesticide exposure for human health and the ecosystem. It also became clear during the workshop that the context of pesticide use was complex and related to poverty, poor education, insufficient housing, and changing climate, to name just a few factors. Multisectoral approaches are needed to improve the situation with many co-benefits for the public health situation. Key are long term strategies, such as capacity building for regulatory agencies, supported by evidence-based policies; continuing education throughout the pesticide supply chain and, ultimately, to farmers; access to actionable science concerning pesticide risk management; the establishment of effective risk communication networks that translate science for end users; viable strategies for scaling up education and communication; an underpinning of consultation, participation, and continuing engagement with end users to determine needs; tracking the status and trends in both adverse impacts and the benefits of interventions.

This workshop report should not be considered a consensus report. Discussions were stimulated with presentations and polling questions. However, the online format did not allow for in-depth discussions. It should also be noted that only a minority of the registered participants voted. At the beginning of the workshop, we also tried a Delphi approach and repeated the polls after the discussion. This resulted only in minor changes and was, thus, not presented in this paper. This may reflect that the discussion time was short and not context specific. Further, it indicates that the participating experts had an established opinion, which may depend on their specific background. Further discussion would be needed to understand different judgments and to derive more specific interventions for various contexts. 

The workshop focused mainly on classical agricultural areas and had less focus on countries, such the Congo or Cameroon, that had many forests that were converted to agricultural lands. Fewer data are available on the effects of pesticides on the ecology in such frontier environments.

## 5. Conclusions

In conclusion, interventions to reduce pesticide use are still not well explored and researched. This workshop calls for more participatory actions and research to improve the situation, in particular for smallholder farmers and surrounding areas to protect their health and well-being. The key areas of improvement identified were training on pesticide usage techniques, research on the effectiveness of interventions targeted at exposure reduction and/or behavioral changes, awareness raising and implementation of adequate policies, and enforcement of regulations and processes. Future research should follow a research translational paradigm, including affected communities, and search for risk reduction co-benefits such as access to health care, dealing with the pandemic, or waste management.

## Figures and Tables

**Figure 1 ijerph-19-08973-f001:**
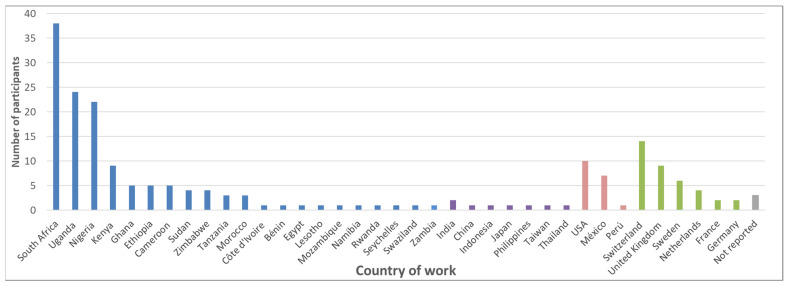
Country of work of the registered workshop participants (blue = Africa; purple = Asia; red = Americas; green = Europe).

**Figure 2 ijerph-19-08973-f002:**
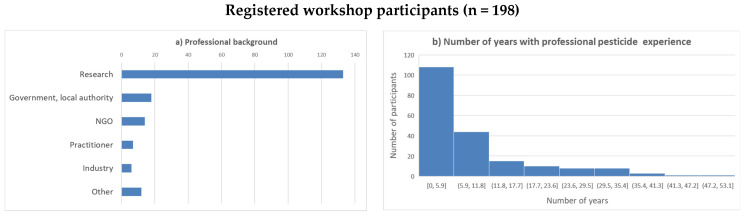
Registered workshop participants: (**a**) professional background; (**b**) number of years of professional pesticide experience.

**Figure 3 ijerph-19-08973-f003:**
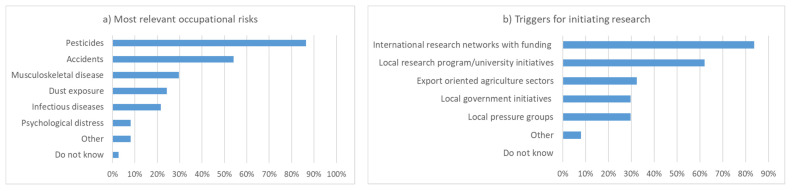
(**a**) The most relevant occupational risks of pesticide applicators; (**b**) the most relevant triggers for initiating research as assessed by the workshop participants (n = 37, up to 3 responses per person).

**Figure 4 ijerph-19-08973-f004:**
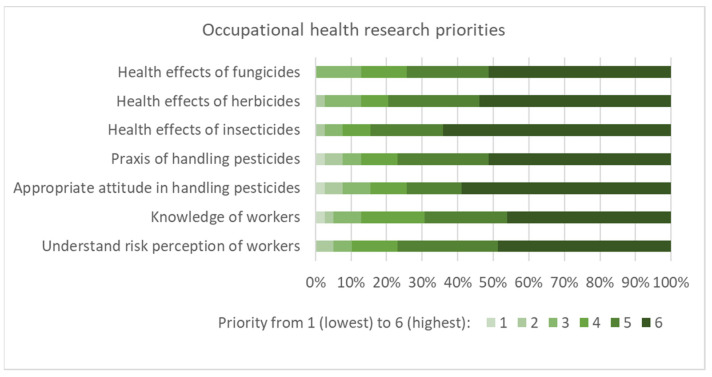
Proposed occupational health priorities for research as assessed by the workshop participants (n = 39, priority rated from 1 to 6, denoting lowest to highest).

**Figure 5 ijerph-19-08973-f005:**
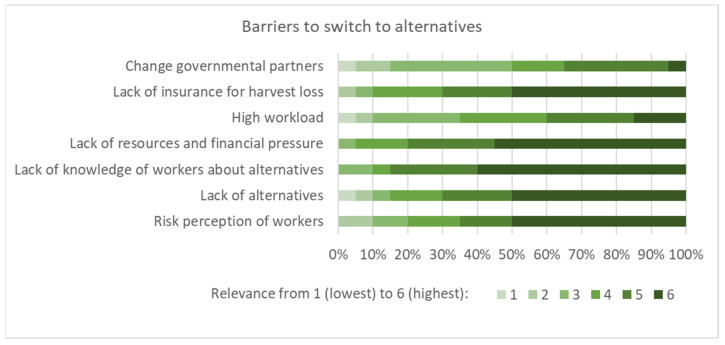
Potential barriers to switching to alternatives as assessed by the workshop participants (n = 20, importance rated from 1 to 6, denoting lowest to highest).

**Figure 6 ijerph-19-08973-f006:**
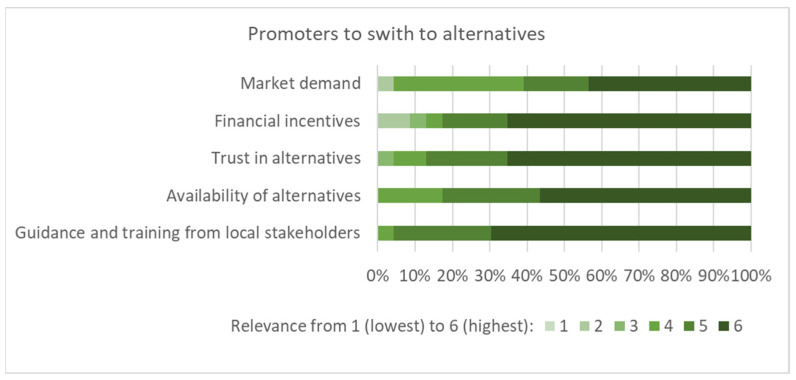
Factors that may promote switching to alternatives as assessed by the workshop participants (n = 23, importance rated from 1 to 6, denoting lowest to highest).

**Figure 7 ijerph-19-08973-f007:**
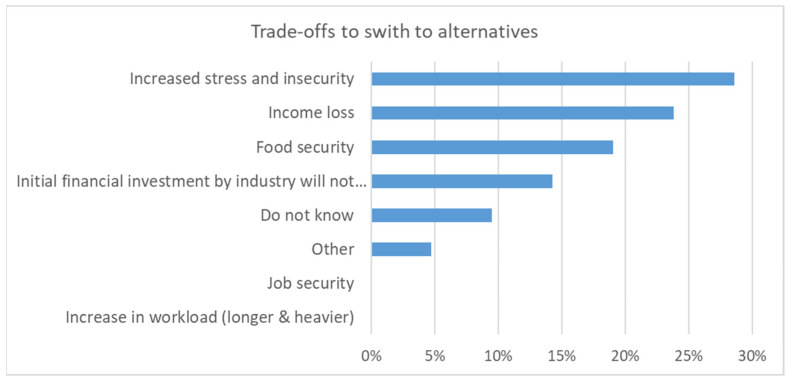
Trade-offs to switching to alternatives as assessed by the workshop participants (n = 23).

**Figure 8 ijerph-19-08973-f008:**
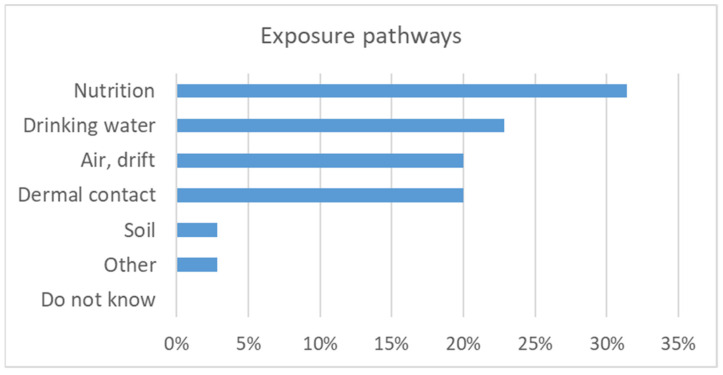
Most relevant exposure pathways as assessed by the workshop participants (n = 35).

**Figure 9 ijerph-19-08973-f009:**
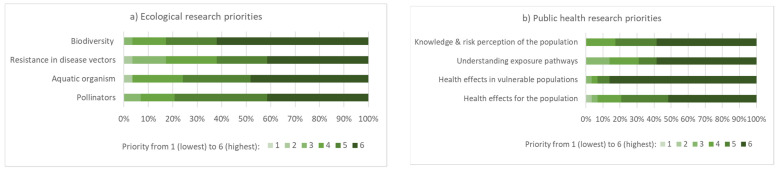
Most relevant **a**) ecological and **b**) public health research priorities as assessed by the workshop participants (n = 29, relevance rated from 1 to 6, denoting lowest to highest).

**Figure 10 ijerph-19-08973-f010:**
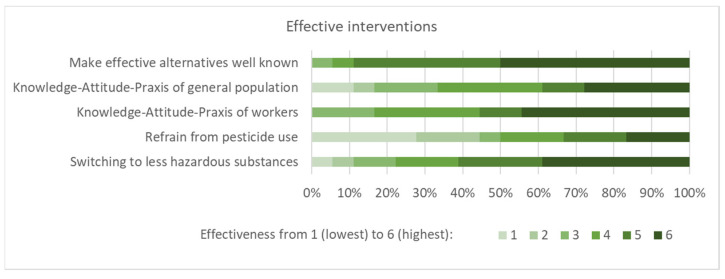
Rating of the question: which interventions are known to be effective for reducing pesticide exposure in the environment? (n = 18, effectiveness rated from 1 to 6, denoting lowest to highest).

**Figure 11 ijerph-19-08973-f011:**
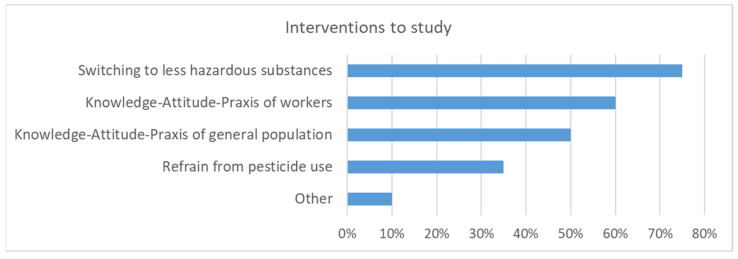
The effectiveness of which interventions that should be addressed in research as assessed by the workshop participants (n = 20, multiple-choice).

**Figure 12 ijerph-19-08973-f012:**
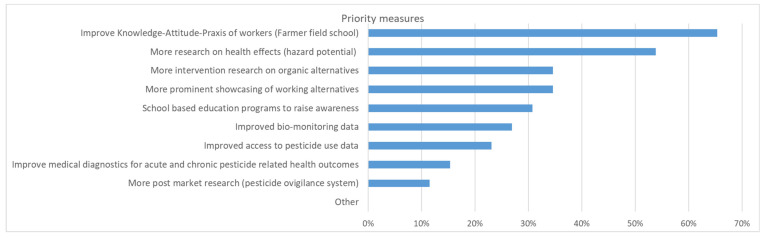
Which priority measures should be implemented (n = 26, up to 3 responses per person).

**Figure 13 ijerph-19-08973-f013:**
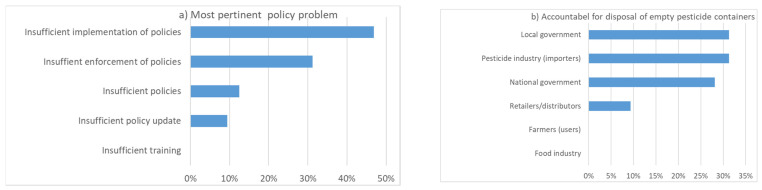
(**a**) What is the most pertinent problem when it comes to policy; (**b**) who should be most accountable for the management/disposal of empty pesticide containers (n = 32).

## Data Availability

Presentations of the workshop and recordings of the talks are available at: https://www.swisstph.ch/en/about/events/interventions-to-reduce-pesticide-exposure-from-agriculture-sector-in-africa/ accessed on 18 July 2022.

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
