# Peer review of "Interventions to Reduce Pesticide Exposure from the Agricultural Sector in Africa: A Workshop Report"

_ijerph, 2022, doi:10.3390/ijerph19158973_

Round 1
Reviewer 1 Report
Dear Authors,
the results presented are important and should be published. However, my concerns are the following:
All:
- The numbers of the lines are missing.
Abstract:
- The importance of this study should be emphasized.
- The main conclusion should be added in the end.
Introduction:
- The main pesticide and its adverse health effects should be explained in more detail.
Materials and Methods:
- The identification of the articles should explain in more detail.
- The methods for obtaining the results that are shown in the figures 3-13 should be better explained.
Results:
- The main conclusions should be added at the end of each section.
- The tables of the found literature and the most important data should be added for more illustrative results.
- Page 7: A space within a paragraph should be delated.
- Not all figures were discussed in the text. Figure 11 is missing.
Discussion and conclusion:
- The last section should be entitled just “Conclusion”. The discussion should be placed in the previous text.
- The main conclusions should be emphasized more clearly and shortly.
Overall, mine suggestion is that the manuscript would be acceptable with major revision. Substantial changes should be carried out before the acceptance.
Good luck!
Reviewer 2 Report
I thoroughly enjoyed reading this work. Despite a large number of authors, the text is consistent, clear, and readable. The language standard is very high, the work has a proper, scientific tone. The authors take up an important issue and do so in a cross-cutting and complete manner. The scale of the efforts made is quite impressive. The aim of the paper and its contribution to the common knowledge are well specified in the introduction chapter. The conclusions presented were consistent with the rest of the paper and based on the results. The weaker side of the work is graphic design. The figures are aesthetically pleasing and legible, however, some of them have been merged into pairs with not entirely common content. Some graphics need to be presented separately. The numbers of the figures will change and the references in the text must follow them. There is a large number of self-citations making it difficult to assess the appropriateness of all of them. I understand that with this number of authors and their involvement in the topic, this is inevitable and must be accepted. The review used the most recent scientific papers. Most of the papers cited (54%) were published in the last 5 years, including eight papers published in 2022. About 23% of the papers were from the last 10 years, 16% were older papers. For 6 items, mostly websites, no year of publication or access date was provided. All items from the references were used at least once in the text. The paper has been provided in a PDF version. The number of lines in the document was not shown which made the review significantly more difficult. This will also make it more difficult for the authors to address their responses. A Word file would be preferred in this situation. The document should be accepted for publication after minor corrections.
Specific comments:
Abstract:
Reduce the word count to 200 words.
Keywords:
Reduce the number of keywords to 5.
Introduction:
There is an unnecessary wide margin in the second paragraph, this should be corrected to save space. Apply throughout the document.
Methods:
Figure 1. Why is Thailand represented by a different color than the rest of Asia?
Why is Peru represented by a color like the European countries?
Figure 2. The plot is not visible in its entirety. Figure 2 should be separated into two because the graphics show completely different data. As a result, the order of the figures will change, which will need to be reflected in the text.
Results:
Occasionally there are elements of text in red. Please use the standard color.
When naming specific species, it is useful to also include the Latin name, e.g.: fall armyworm Spodoptera frugiperda
Figure 3 Figure 3 should be split into two separate ones.
Figure 12. Figures should have headers.
Figure 13. These figures should be separated.
References:
In the case of weblinks, please provide the access date. This applies for example to references no. 2, 28, 29, 30, 51, and 69.
Reviewer 3 Report
The authors present a summary of workshop held in November 2021 focused on pesticide use in agricultural practices in Africa. The workshop summary includes a description of the problems and knowledge gaps related to pesticide use in Africa, a summary of how the workshop was organized, and detailed summaries of many of the topics that were discussed during the conference.
Overall, the workshop summary is effective at communicating the information discussed during the workshop and would be valuable to any researchers interested in this topic that were unable to participate in the conference. Therefore, it is recommend that the manuscript be accepted for publication.
Specific Comments:Figure 2: It would be clearer to the reader to label the two graphs in figure 2 as A and B or separate the graphs into separate figures. The title can be removed from the second graph (“Number of years with professional pesticide experience”) can be removed.
Page 6: The citation “Sapbamrer et al 20202 [35]” should be changed to “Sapbamrer et al. [35]”
Round 2
Reviewer 1 Report
Dear Authors,
substantial changes were made. In my opinion the manuscript is now acceptable for the publication.
Kind regards!